# Kantian Ethics and the Animal Turn. On the Contemporary Defence of Kant’s Indirect Duty View

**DOI:** 10.3390/ani11020512

**Published:** 2021-02-16

**Authors:** Samuel Camenzind

**Affiliations:** Unit of Ethics and Human-Animal Studies, Messerli Research Institute, University of Veterinary Medicine, Vienna, Medical University of Vienna, University of Vienna, 1210 Vienna, Austria; samuel.camenzind@vetmeduni.ac.at

**Keywords:** animal ethics, animal turn, Kantian ethics, duties regarding animals, indirect duty view, animal rights

## Abstract

**Simple Summary:**

Criticism of Kant’s position on our moral relationship with animals dates back to the work of Arthur Schopenhauer and Leonard Nelson. Against this, Kantians have continued to defend Kant’s view that animals lack a moral status. Kant’s contemporary defenders have, however, highlighted extensive practical consequences for the protection of animals in favor of Kant’s position. This paper explores the argument from these extensive duties to animals in Kant’s ethics and seeks to show that Kantians underestimate essential differences between Kant and his rivals today (including proponents of animal rights and utilitarians) on both a practical and fundamental level. It also argues that those defending Kant tend to neglect theory-immanent problems in Kant’s ethics arising from unfounded value assumptions and unconvincing arguments for the denial of animal moral status. It is suggested that although the human-animal relationship was not a central concern of Kant’s, examination of the animal question within the framework of Kant’s ethics helps us to gain conceptual clarity about his duty concept and the limitations of the reciprocity argument, i.e., the notion that morality is a system of reciprocal relationships.

**Abstract:**

Criticism of Kant’s position on our moral relationship with animals dates back to the work of Arthur Schopenhauer and Leonard Nelson, but historically Kantian scholars have shown limited interest in the human-animal relationship as such. This situation changed in the mid-1990s with the arrival of several publications arguing for the direct moral considerability of animals within the Kantian ethical framework. Against this, another contemporary Kantian approach has continued to defend Kant’s indirect duty view. In this approach it is argued, first, that it is impossible to establish direct duties to animals, and second, that this is also unnecessary because the Kantian notion that we have indirect duties to animals has far-reaching practical consequences and is to that extent adequate. This paper explores the argument of the far-reaching duties regarding animals in Kant’s ethics and seeks to show that Kantians underestimate essential differences between Kant and his rivals today (i.e., proponents of animal rights and utilitarians) on a practical and fundamental level. It also argues that Kant’s indirect duty view has not been defended convincingly: the defence tends to neglect theory-immanent problems in Kant’s ethics connected with unfounded value assumptions and unconvincing arguments for the denial of animals’ moral status. However, it is suggested that although the human-animal relationship was not a central concern of Kant’s, examination of the animal question within the framework of Kant’s ethics helps us to develop conceptual clarity about his duty concept and the limitations of the reciprocity argument.

## 1. Recent Developments in Kantian Ethics Regarding the Animal Question

To begin with a terminological note, I will introduce the distinction made by Onora O’Neill [1] between Kant’s ethics and Kantian ethics (see also [2]) (p. 1ff.). “Kant’s ethics” refers to Kant’s own moral philosophy, developed in the Groundwork of the Metaphysics of Morals (1785), Critique of Practical Reason (1788), The Metaphysics of Morals (1797), and the lecture notes to his moral philosophy. “Kantian ethics” refers to approaches to ethics which are informed by Kant’s views but modify them in different degrees. These modifications concern also the moral standing of animals.

Historically, Kantians have shown little interest in the status of the human-animal relationship in Kant’s ethics and the view that Kant classified animals as things has been implicitly accepted. A few papers dealing with Kant’s views on animals were published in the 1970s and early 1980s (e.g., [3,4,5,6,7]), but Kant scholars have not addressed the animal question with real enthusiasm until recently.

This is interesting because the critique of Kant’s views on the moral standing of animals can be dated back to Arthur Schopenhauer (1840) [8], Albert Schweitzer (1923) [9], and Leonard Nelson (1932) [10]. Moreover, in recent history various pieces of animal welfare legislation have not only illustrated a paradigm shift, moving from an indirect, anthropocentric perspective on animal protection to a direct, sentientist one (cf. [11] p. 63ff.), but also declared that animals are, legally speaking, no longer things. This legislation implies that the distinction between persons and things, originating in Roman law, is outdated.

Looking at the literature, we can see that the discourse changed significantly in the mid-1990s with the work of notable scholars like Paul Guyer (1993) [12], Christine Korsgaard (1996) [13], Allen W. Wood (1998) [14], and Onora O’Neill (1998) [15]. From this point onwards, Kant’s position on the human-animal relationship came into the foreground. It came to be referred to regularly in dissertations (e.g., [16,17,18]) and even introductions to Kant’s moral philosophy [19]. Some commentators now treated it as one of Kant’s major mistakes, for example [20] (p. 210f.), [21] (p. 22), [22] (p. 175). These developments have to be seen in the wider context of what has been called the “animal turn” in the humanities. Inspired by the term “linguistic turn”, the “animal turn” [23] refers to an intensification of interest in animal issues in the humanities and social sciences.

The animal question is important for Kantians, because when we track animals in Kant’s ethics, we arrive at key questions about Kant’s axiology, concept of duty, and view of the foundation of ethics. On the other hand, the study of Kant’s ethics is likely to be fruitful for animal ethicists, given the impact of Kant’s categories and terminology on the field of animal ethics. A Kantian legacy can be detected, for example, in the division of duties into direct and indirect sorts [10] (p. 137ff.), [24] (p. 150ff.), in references to the absolute moral value and dignity of animals, and in the prohibition of excessive instrumentalization of animals [24] (pp. 235, 248f.), which, by 2005, had even been built into the Swiss Animal Protection Act (See Article 3, litera a), which states, that if the excessive instrumentalization of the animal cannot be justified by overriding interests, this constitutes a disregard for the animal’s dignity (for recent debate see: [11,18,25]).

However, as Onora O’Neill once stated: “Although claims on behalf of non-rational beings are easy to state, they are hard to establish” [15] (p. 220). Indeed they are, and one might add that the moral status of animals is a particular problem for Kantians. According to John Rawls cruelty to animals is wrong, but he denies that animals are subjects of justice in his Theory of Justice (1971) [26] (p. 512). Jürgen Habermas shares the moral intuition that we are morally obligated to sentient animals as “vulnerable creatures” [27] (p. 106), but he also believes that the asymmetric conditions of communication in humans and animals (and plants) do not allow us to regard animals as subjects of morality [27] (p. 111). (In this paper, it will not be possible to discuss the approaches developed to fill the gaps in Rawls’ and Habermas’ treatments of the moral status of animals. Several contemporary works discuss this issue [28,29,30,31,32]).

In recent work on Kantian moral philosophy, three kinds of strategy have been adopted in reaction to Kant’s denial of moral status to animals. (a) The theory-immanent approach retains the main pillars of Kantian ethics. It argues that Kant was mistaken in excluding non-rational beings from the moral community, and that his ethical framework can accommodate an improved position on the moral standing of animals (e.g., [14,21,22]). (b) A second theory-transcendent approach modifies Kant’s ethics significantly by adding to the central concepts of autonomy and duty some additional concepts borrowed from non-Kantian theories [13,24,33,34]. Because theory-immanent and theory-transcendent strategies introduce major revisions of Kant’s ethics and sometimes twist central components of Kant’s ethics beyond recognition [35] (p. 138), (c) a third Kantian approach goes back to square one and defends Kant’s indirect account of our obligations to animals. Here it is argued that it is not only impossible, but also unnecessary, to establish direct duties to animals, because the Kantian notion that we have indirect duties to them has far-reaching practical consequences and is to that extent adequate. Indeed, when they are considered in their historical context, Kant’s writings on this topic can be seen as progressive (e.g., [15,16,20,36,37,38]).

I wish to explore this third approach and examine the arguments for Kant’s indirect duty view, focusing especially on the allegedly far-reaching practical consequences. In Section 2, I present Kant’s view and the arguments supporting his ethics. Then, in Section 3, I compare these with the common-sense view of morality, utilitarianism, and the animal rights theory on both a practical level and fundamental level.

## 2. In Defense of Kant’s Indirect Duty View

Basaglia summarized several reasons why Kant’s ethics is attractive for contemporary Kantians (cf. [38]) (pp. 1725–1728). Among them are the strictness of our Kantian duties to animals (a point further explored below) and the immunity of Kant’s ethics to two difficulties besetting utilitarianism and Schopenhauer’s ethics of compassion: first, the problem that the interests of the majority will trump those of the minority; and second, the “universality” problem that compassion cannot be the ground for a stable system of morality because the capacity for it is not distributed equally and universally within humankind).

Those who favor what I have called the third response to Kant’s denial of moral status to animals and defend an indirect duty view also highlight the far-reaching practical consequences of Kant’s position read in this way. In her conservative account, Lara Denis notes that Kant goes even further than the utilitarian would by formulating duties to insects (cf. [36]) (pp. 418f.). It has to be mentioned, that Denis’ discussion was published in 2000. Since then, new scientific findings on the sentience of insects have given renewed impetus to the discussion (e.g., [39,40,41]). This means that if science can provide compelling reasons for believing that insects are sentient, utilitarians will recognize them as morally relevant entities, leaving Denis’s argument outdated. Matthew Altman argues, in the same vein, that Kant’s restrictions on animal use are consistent with non-anthropocentric positions in environmental ethics (cf. [42]) (p. 16). He compares Kant’s objections to the mistreatment of animals with the criticism voiced by Peter Singer’s preference utilitarianism and Tom Regan’s animal rights view. Discussing the killing of animals for trivial culinary pleasure, the handling of working animals, hunting for sport, lifeboat scenarios, and animal experimentation, Altman concludes that “the implications of Kant’s position here do not contradict animal rights and animal welfare theories” [42] (p. 20). Similarly, Patrick Kain remarks, “[…] it is not clear that Kant’s theory demands, at a fundamental level, much less regard for non-human animals than many of its rivals do” [20] (p. 232).

Indeed, in their historical context Kant’s writings on the human treatment of animals can be seen as progressive. On the few pages where he mentions them—in a section titled “On Duties to Animals and Spirits” in ‘his’ Lectures on Ethics, and in “Episodic Section. On an Amphiboly in Moral Concepts of Reflection, Taking What is Man’s Duty to Himself for a Duty to Other Beings” in the Metaphysics of Morals (1798)—he covers a wide range of human-animal relationships and mentions several, rather extensive, duties we have regarding (in Ansehung) animals [43] (p. 443). (I hesitate to call the Lectures ‘his’ work because the Collegium Philosophiae practicae universalis una cum Ethica consists of various student notes and transcripts of the lecture courses Kant taught until the winter term 1793/94, and these have been handed down and edited with differing degrees of care. Because they are not authorized by Kant—who was quite skeptical about student notes—one should be careful with interpretations that rely on literal readings. The Lectures nevertheless serve as valuable supplements to Kant’s own writing. According to my comparison, the notes “on duties to animals and spirits” correspond very closely with each other, and more importantly for my argument in this section the structure of the lecture is preserved (cf. [44]) (p. 327)).

In today’s language, the areas here would be called livestock consumption, animal research, pet keeping, and wildlife. These duties do not merely mirror Western, common-sense moral attitudes to animal treatment. They go beyond this, as the duties are both negative and positive. According to Kant we are only permitted to slaughter animals painlessly; the “violent and cruel treatment of animals” [43] (p. 443) is forbidden. We are permitted to use work animals as a source of power, but we must not use them beyond their capacities. With his prohibition of animal use in “agonizing physical experiments for the sake of mere speculation” [43] (p. 433) or:‘in sport’ (*als ein Spiel*) [45] (p. 460), when alternatives exist to reach the same end, Kant is in agreement with the so-called 3Rs principles (see [46]) (p. 1771). The 3Rs principles states that an experiment involving animals is morally (and often also legally) permissible only if: No alternative method using no animals exists (replacement); no more animals are used than necessary (reduction); and as little harm as possible is inflicted on the animals (refinement) (Russell and Burch 1959) [47]). Kant also praises Gottfried Wilhelm Leibnitz, who, when he observed and handled a little worm, put it gently back on a leaf so that it did not come to harm [45] (p. 460), (see also [48]) (p. 160). Again, in endorsing the notion that owners should be grateful to their horses, or their dogs, for their long service as if they were members of the household, Kant goes beyond any current welfare law.

In defense of Kant, one should further mention that in the Metaphysics of Morals animals are protected by the highest category of duty—the perfect duties to oneself as a moral being. This strengthens the significance of the human-animal relationship and means that Kant’s position in the Metaphysics of Morals must be differentiated from other indirect duty views such as classical contractarianism and various forms of anthropocentrism (although, as regards contractarianism, it should be mentioned that new approaches have recently been developed which argue for the moral status of animals within the contractarian approach (e.g., [49,50]).

According to Kant’s moral argument against cruelty, torturing animals is impermissible even if the animal is not protected by its status as the property of its owner, and even if no third party would be negatively affected. As a consequence, Kant’s position is immune from certain criticisms that have been raised against other indirect duty views. For example, the complaint that Kant’s argument is based “on fragile empirical claims about human psychology” [28] (p. 330) or a “dubious psychological claim” [51] (p. 2) only concerns his Lecture, not his critical writings (cf. [52]). The reach of these duties, and their standing within the system of duties, explain why some Kantians argue that animals are protected sufficiently in Kant’s framework.

## 3. Critique of the Contemporary Defense

I will now argue that although the proposed human-animal-relationship conforms with what is, prima facie, contemporary common-sense morality, the conclusion that Kant can be compared to today’s animal rights and welfare theorists, or that he even goes beyond utilitarianism, is premature. The defense of Kant’s indirect duty view is not successful, because it (3.1) overestimates the moral reach of Kant’s indirect duty view, (3.2) underestimates the fundamental differences between Kant’s position and direct duty views, and (3.3) fails to provide a satisfactory argument showing why humans are allowed to use animals as mere means in general (e.g., by caging, killing, or harming them). It would be beyond the scope of this paper, and against the intention of the author, to defend an alternative view of the way the moral status of animals can be justified. Nevertheless, the reader will find in-depth enquiries in support of moral status for animals in the following works: Kantianism and animal rights: [10,14,15,17,21,24,33,34,53]; utilitarianism: [54]; contractarianism [49,50]; virtue ethics: [55,56]; and other publications that cannot be safely categorized in these four traditions: [8,9,28,57]. The last point concerns central concepts of Kant’s ethics, his value theory concerning animals and the range of the reciprocity argument.

(3.1) First of all, it must be mentioned that in spite of its historically progressive practical implications, and despite the presentation of it given by its defenders, Kant’s account differs markedly at the practical level from today’s rival accounts. The animal rights view formulated by Regan [24] leads ineluctably to veganism, the total abolition of invasive animal experimentation, and comprehensive dissolution of animal agriculture and recreational hunting, because these practices violate animals’ right to bodily integrity, or their fundamental right to life. In contrast with Kant, Regan’s position outlaws the painless slaughter of animals and any method of drug development or toxicology testing that involves distress for the animals, even if they are ‘employed for a good purpose’ (a factor Kant takes to be relevant [45] (p. 460) (For a deeper exploration of the Kantian perspective on animal experimentation, see [46], where it is argued—against the view Kant’s develops in the Doctrine of Virtue—that agonizing experiments on animals are to be condemned even if they serve a morally required purpose because they violate the perfect duty one has to oneself as a moral being). Therefore, there is a big difference at the practical level between Kant’s indirect duty view and the animal rights theory.

Equally, in the utilitarian tradition, the “principle of equal considerations of interests” requires us to consider and weigh the interests of humans and animals impartially. As Singer famously says, “an interest is an interest, whoever’s interest it may be” [54] (p. 20). Although animal research is not categorically prohibited, even research that is done for the sake of a “good purpose” can be morally forbidden if the harm done to the animals outweighs the resulting human benefits. This means that Federica Basaglia’s claim (cf. [38]) (p. 1726) that the strength of Kant’s position on animal experimentation is similar to that of the utilitarian position Peter Singer advocates is unconvincing. Kant’s conditions of ethically permitted animal experimentation are (a) important benefit, or benefits, for science or society (instrumental necessity) and (b) reduction of the expected harm to the animals involved to the minimum (rather like the 3Rs principles). While utilitarians would agree with Kant on these two requirements, they would not see them as sufficient. In addition, the utilitarian will want to (c) run a harm-benefit analysis. (As a matter of fact, these are nowadays implemented in most animal experimentation regulations in Europe (e.g., [58]). Within Kant’s ethics, the justification of “agonizing physical experiments” does not require this last step of a harm-benefit analysis—which leads to the second, and fundamental, point that needs to be made.

(3.2) In Kant’s ethics, the interests, and the welfare, of animals are not in themselves given moral consideration. This is a categorical difference at the fundamental level. It reminds us that we should not underestimate the important differences between Kant and his rivals today, nor the rather different outlooks expressed in Kant’s views and those of contemporary common-sense morality. For Kant, all duties concerning animals belong “*indirectly* to a human being’s duty *with regard to* these animals; considered as a direct duty, however, it is always only a duty of the human being to himself” [43] (p. 443). Thus in Kant’s view, when it is stated precisely, one’s duties regarding animals are related to the duty to oneself as a moral being, to preserve one’s shared feeling (Mitgefühl), which is “a natural predisposition that is very serviceable to morality” [43] (p. 443), but this shared feeling does not, in itself, have moral value. Because non-human animals, as an “analogue of humanity” [45] (p. 459), are in many respects similar to rational animals, it is easy to confuse the object of a duty with the source and authority of that duty—the being to whom we are obligated. This is why Kant titles this section “Amphiboly in Moral Concepts of Reflection” [43] (p. 442).

Consider Altman’s statement that “human interests do not automatically trump animal interests” [42] (p. 19). From the perspective of Kant’s indirect duty view, this is misleading, because animal interests do not count morally at all. For the same reason, the assumed advantage of Kant’s moral philosophy over other accounts accruing from the fact that it includes invertebrates in the moral sphere is unfounded. Unlike biocentrists such as Albert Schweitzer [9] or Paul W. Taylor [57], Kant does not respect the Leibnitz’s worm in a moral way. The use of animals is (in some ways, at least) strongly restricted by Kant with reference, not to the interests of the animals, but to duties to oneself (Metaphysics of Morals) and to duties to others (Lectures). In other words, because animals are not beings which can be respected (in a moral sense), their interests, feelings, and needs are not morally considerable, and therefore they can neither trump nor be weighed against human interests. As Kant clearly states: “*Respect* is always directed only to persons, never to things. The latter can awaken in us *inclination* and even *love* if they are animals (e.g., horses, dogs, and so forth), or *fear,* like a sea, a volcano, a beast of prey, but never *respect*” [48] (p. 76). There is a strong contrast between these remarks, on the one hand, and the animal rights framework, utilitarianism, and common-sense morality on the other. It is precisely the idea that the slaughter of a veal calf purely for culinary pleasure, or the experimental use of primates in research causing them severe, inescapable suffering, or the burning of a living cat, do not count morally that is, for some philosophers, simply incomprehensible. In Regan’s rights view, all sentient beings possess inherent value, and therefore all have the very same moral right to respectful treatment (cf. [24]) (p. 279). In utilitarianism, where the interests of a minority of animals can be sacrificed in the interests of a human majority, their interests still count morally and are weighed impartially. Even in the common-sense animal welfare view, which protects animals minimally and only prohibits the imposition upon them of unnecessary suffering, the interests of the animals are morally considered. This does not mean that these ethical positions are superior to Kant’s own in respect of their justificatory base. However, it merely shows that, at a fundamental level, the differences here are immense. A last point should be added here.

Against the moral considerability of animals, Kant follows the termini technici of Roman law and distinguishes sharply between persons and things (cf. [59]) (p. 428). Persons are defined as morally autonomous beings with the faculty of personality and the ability to act morally in consequence. They bear absolute, inner worth, and therefore dignity, which makes them the subject of moral respect (cf. [59]) (p. 428, 435, 436). Animals, by contrast, are excluded from the moral community. Together with minerals, plants, and all other non-autonomous beings, they count as “things”. According to Kant, they possess only a conditional, relative worth for persons (cf. [59]) (p. 428). In other words, animals are “means and instruments to be used at will for the attainment of whatever ends he [man] pleased” [60] (p. 225), or, as Kant puts it elsewhere, things “with which one can do as one likes” [61] (p. 127). The (legal and moral) property status of animals is the next point at which Kant’s view is incompatible with animal rights views—as, for example, Gary Francione argues very convincingly in Animals, Property and the Law (1995) [53]. In this context it has to be mentioned, that the legislatures in Austria (1988), Germany (1990), and Switzerland (2003) have decreed that animals are, nominally, not things anymore (corresponding articles were laid down in Austria in the Allgemeinen Bürgerlichen Gesetzbuch (ABGB § 285a) in 1988, in Germany in the Bürgerlichen Gesetzbuch (§ 90a BGB) in 1990, and in Switzerland in the Zivilgesetzbuch (Art. 641a Abs. 1 ZGB) in 2003). In other words, the strict distinction between persons and things is not self-evident—especially because it dates back to Roman law, where the category of things included not only animals, plants, and non-living entities, but also children and slaves. Therefore, the distinction, as a culturally and historically contingent fact, should not be regarded as self-evidently sound, nor should Kant’s modification of it be taken for granted, because here the dominant distinction in Roman law is dissolved, although this paradigm change has not yet altered the situation of animals much in practice [62] (p. 146). In the end, the property status of animals shows once more that an unbridgeable gap exists between Kant’s ethics, on one side, and the animal rights view and common-sense perceptions of animals today, on the other—at least, in the German-speaking countries of Europe.

The comparison of Kant’s view on animals with the animal rights view and utilitarianism demonstrates clearly how big the difference between these two positions, at both a practical and a fundamental level, are. Although Kant mirrors contemporary Western, common-sense morality of animal treatment occasionally, or even goes beyond it, the reason why we should protect animals differs here too.

(3.3) Finally, I want to mention a further point at which I think the contemporary defense of Kant’s indirect duty view should be challenged. Even if philosophers have good reasons to prefer Kant’s ethics and Kantianism over utilitarianism, contractarianism, virtue ethics, and other approaches, when it comes to the human-animal relationship and its underlying axiology, it remains true that Kant’s position is not only at odds with twenty-first century science, but also insufficiently supported by argument. It will not be possible here to discuss this topic in depth, and to offer an alternative view, but my aim is to show that Kant’s position on the human-animal relationship should not be taken for granted and simply assumed to be sound.

The twin premise that animals “have value only relative to human needs and desires, or relative to the ends that human being freely set for themselves”, followed by the conclusion that animals can be treated merely as means to human ends (Altman [37] following Kant), is problematic for several reasons.

From a scientific perspective, in a post-Darwinian world it is clear that human beings (*Homo sapiens*) are, like every other species, an arbitrary product of evolution. According to the theory of evolution, no scala nature, with humans on top and all other species below, exists. The teleological view Kant presents in the Conjectures on the Beginnings of Human History [60] (p. 114), which sees the sheepskin, or any other body-part of the sheep (*Ovis gmelini aries*), or, analogically, the milk of cattle (*Bos taurus*), as something evolved for the sake of the human species, is, from a scientific point of view, unworthy of discussion. Animals live their own lives, maintain and reproduce themselves independently of human beings, and have their own communities and individual welfare needs that matter to them. That is why Taylor describes all flourishing beings as “teleological centers of life” [57] (p. 119), and why Korsgaard claims that “[o]ur own moral standards demand that all sentient beings should have comfortable and happy lives […]. And those standards, I believe, demand that we treat all animals with compassion and respect, as ends in themselves, not as mere means to our own ends” [63]. To sum up, it is prima facie odd to assume that animals are here on earth to serve human ends as Kant assumes.

Kantians may respond that Kant argues, not at a scientific level, but a normative one. He does not present a description of the world. He offers a normative picture of the way in which the world should be. What are the arguments for the conclusion that animals are means to human ends, and that animals have only relative value for rational beings?

Unfortunately, for Kant’s sympathizers there are only a few passages in his oeuvre where he presents an explicit argument against the moral status of animals. As Franklin states [21] (p. 40), a kind of dogmatism about the human-animal divide appears throughout Kant’s work: Kant simply assumes that animals are non-rational creatures, and that therefore they do not deserve moral respect. This is largely true, but there are exceptions. In Groundwork, Kant introduces a regress argument (also called the “argument by elimination”, (cf. [22]) (p. 118)), in his derivation of the formula of humanity (cf. [59]) (pp. 427–429), against the moral status of animals. In the course of this argument, he excludes three potential possessors of absolute worth: relative ends that are grounded in inclinations, inclinations themselves, and animals (whose actions arise from inclination only). With the focus on inclinations in this argumentum ad regressum, it does not make sense to mention other “things”, like plants or inanimate objects, and contrast them with “persons”. For Kant, it seems to have been entirely obvious that the former do not deserve to be regarded as ends in themselves with inherent worth—against this view, see the contributions of the biocentrist Paul W. Taylor [57] or the holist Martin Gorke [64]. According to the argument, animals are merely subjective ends. They have value for persons, “whose existence in itself is an end” [59] (p. 428). Although Franklin overlooks Kant’s argument, I share Franklin’s position: Kant’s conclusion that animals are not ends in themselves, but mere means with only relative value for persons, is unconvincing. Kant provides no further explanation or justification for his conclusion, and it is widely agreed among Kantians that Kant’s axiology is rather more presupposed than it is properly grounded (cf. [65]) (p. 142), [66] (p. 147), [21] (p. 40), [15] (p. 213ff.). Even if Kant is right that persons with the capacity for autonomy are ends in themselves with absolute worth, it does not follow that everything else has only relative value for persons. However, this mantra is exactly what he repeatedly states throughout his work (e.g., [45] (p. 459), [60] (p. 114), [48] (p. 87). Animals, considered as “experiencing subjects of a life”, can at least be regarded as beings with independent experiences and interests that are not necessarily linked to persons, because they have “[…] individual welfare in the sense that their experiential life fares well or ill for them, logically independently of their utility for others and logically independently of their being the object of anyone else’s interests’ [24] (p. 243); see also [33,34,63,67,68].

Moreover, (1) “the argument of regression”, two other arguments for the view that animals are not morally considerable can be found in Kant’s writing: (2) “the argument from self-consciousness as differentia specific”’ and (3) “the argument of reciprocity”. Because the former would lead to complex ethological inquiries (e.g., [69,70]) that are beyond the scope of this article, I will focus on reciprocity.

(2) Only one thing has to be noted: Whatever differentia specifica is mentioned by Kant—self-consciousness, rationality, or moral autonomy (and this last is the most precise)—it is an open question whether this has, inevitably, to result in a relationship of domination where animals are no more regarded “as fellow creatures, but as means and instruments to be used at will for the attainment of whatever ends he [man] pleased.” [60] (p. 114). Instead of delivering a “prerogative which, by his [man’s] nature, he enjoyed over all the animals” [ibid.], the criterion could lead to a relation of compassionate care for other creatures. Alternatively, a third relation would be an abolitionism, i.e., the notion that human beings should not interact with other animals at all. This is not the place to argue for one or other of these positions, but this consideration should make it clear that Kant’s vision of domination is neither the only human-animal relationship one can think of nor self-evident.

(3) The essential idea of the reciprocity argument is that morality consists of reciprocal relations of obligation that obtain between persons. Every (human) autonomous being is, at the same time, a legislator (Oberhaupt, auctor obligationis), as well as a subordinate (Glied, subjectum obligationsi), of the moral law (cf. [59] (p. 434f.), [43] (p. 417)). This results in symmetrical relations between persons, and thus in the “systematic union of several rational beings through common laws” [59] (p. 433); [43] (p. 462), i.e., the “kingdom of ends”. Where the argument from reciprocity and Kant’s duty concept is concerned, one can only argue against the fact that animals can obligate us. Because animals are not morally autonomous beings, they cannot participate within the kingdom of ends as equals. Animals cannot be obligated and they cannot be held responsible for their actions.

However, it is impossible to draw any conclusions from this about animals’ value, or about the right way to treat them. This can be illustrated with the analogy of “passive citizens” in Kant’s political philosophy (cf. [43]) (p. 314f.). Although passive citizens, such as women, children, and apprentice house servants, lack “civil personality” and do not participate (for whatever reason) actively in legislation, they are still protected by the law made by active citizens (if one follows Kant and his duty concept, the consequences are challenging: active citizens are not obligated to women, children, and servants, because they are not the authority for a duty, but rather objects of duties among active citizens. For his views on women and servants, Kant deserves criticism. In particular, it must be asked why they are excluded from the legislative process—because, in contrast with children, they meet the criteria of reasoning and participation in legislation).

From the mere fact that active citizens are in a reciprocal relationship with each other, it cannot be concluded that passive citizens have only instrumental value for active citizens; nor can we infer that active citizens have the right to treat passive citizens as mere means. From reciprocity arguments, in other words, we can only derive the conclusion that a specific relationship exists between active citizens (where political philosophy is concerned), or that a specific relationship obtains between autonomous beings (where ethics is concerned). About the relationships with other beings here—with passive citizens or with animals—it is impossible to draw further conclusions, or to establish a value theory.

This is often overlooked by Kantians, because autonomy—the crux and the cornerstone of Kant’s moral philosophy—has a minimum of three different functions, and these are difficult to distinguish and easily confused. First, it is autonomy that provides the groundwork for morality [59] (p. 439f.). Second, autonomy is the reason why rational beings have absolute inner worth and therefore possess dignity that has to be respected (cf. [59]) (p. 435). Kant states that the “dignity of humanity consists just in this capacity to be universally legislating, if with the proviso of also being itself subject to precisely this legislation” [59] (p. 440). Hence, third, autonomy is a condition of being an equal member of the kingdom of ends.

Therefore, even if Kant’s autonomocentrism is sound—this means that autonomy is the foundation of morality (because it alone has absolute inherent worth), and that the kingdom of ends consists of reciprocal relationships between autonomous beings—Kant needs two additional arguments. He needs to show, first, why every other entity has only instrumental value for autonomous beings, and second, why autonomous beings have a right to treat animals and other non-autonomous beings as mere means.

It has to be acknowledged that Kant does not provide a convincing argument showing why animals (or non-rational beings in general) lack moral status. His arguments penetrate only so far—they only allow him to describe relations between autonomous beings. This means that the Kantians who maintain that Kant’s indirect duty view is sufficient, at a practical level, have no foundation for their view. What we need are better explanations: Why is the strict distinction between persons and things accurate? Why should we accept that animals have merely relative value, and specifically value to persons? Moreover, why should we assume that animals are on earth simply to serve human ends? However, neither the argument from reciprocity nor the regress argument can be utilized successfully here. Regrettably, Kant’s moral reflections focus on morally autonomous beings, but have little to say about our relation to the non-autonomous world.

## 4. Outlook

It must be borne in mind that, although the human-animal relationship was not a central concern of Kant’s, examination of the animal question within the framework of Kant’s ethics leads us to think more carefully about central arguments and concepts in Kant’s moral philosophy, including the regress argument invoked in the development of the formula of humanity, the reciprocity argument, and the relationship between members of the kingdom of ends and the highest duty of all. The interface between Kant’s ethics and contemporary animal ethics encourages us to develop improved conceptual clarity about the nature, and limits, of Kant’s duty concept, and about the focus of, and blind spots in, Kant’s moral philosophy. The “animal question” should not be regarded as a side issue in Kantian moral philosophy (without explicitly mentioning the issue, Cavalieris’ work supports my conclusion that the animal question is related to the core of Kant’s philosophy (cf. [71]) (pp. 48–58)). Instead, it can be used as a methodological tool with which to examine, challenge, and improve Kant’s ethics.

## Data Availability

Not applicable.

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
