# Peer review of "Kantian Ethics and the Animal Turn. On the Contemporary Defence of Kant’s Indirect Duty View"

_animals, 2021, doi:10.3390/ani11020512_

Round 1
Reviewer 1 Report
This paper deals with the problem of whether Kant's animal ethics is adequate, given its foundations and practical implications for animals, or whether it is not and must therefore be amended. Against supporters of the Kantian view that we have indirect moral duties towards animals, the author of this paper argues that Kant's ethics in this regard are insufficiently grounded and, furthermore, that they have practical implications incompatible not only with animal rights theories and utilitarianism, but also with common sense morality and, increasingly, the legal system of several nations.
I believe the author's argument to be sound and commendably succinct.
I have, however, two small comments:
(a) It is quite impossible for me to understand how (as the author says some authors suggest) 'Kant goes even further than the utilitarian would by formulating duties to insects' (106). Whatever Kantians say, if insects (or, more accurately, invertebrates in general) are sentient, then their interests must be factored in the utilitarian calculus. Moreover, more pragmatically, to the extent that we have evidence in support of invertebrate sentience, we are morally required to incorporate the possibility of harming/benefit them in our expected utility calculations. In fact, we do seem to have conclusive evidence of octopus sentience (see Cambridge Declaration on Consciousness, Low et al. 2012), and diverse degree of evidence regarding the sentience of other invertebrate animals. In addition, it would be interesting to discuss to what extent Kant's argument works for animals of whose lack of sentience we can be sure. If the indirect duty view necessitates the sentience of the entities regarding which we have the duties, then non-sentient animals would be excluded. If so, and if most invertebrates turn out not to be sentient, Kant's view and the utilitarian's would agree that we have no duties to them.
(b) Regarding the argument from self-consciousness as differentia specifica, I think it would be gentler to the reader to (i) move the mention of it to a footnote and then (ii) provide a brief description of it.
(c) There is a further problem in applied animal ethics which may help to show the steep practical divergence between Kant's ethics and utilitarianism. This is the problem of wild animal suffering. If, as the evidence suggests, wild animals tend to have lives of net suffering to naturogenic (as opposed to anthropogenic) harms, then we have utilitarian reasons to intervene in nature in order to help them. Even to the point of redesigning nature if necessary. Some have argued that on a modified Kantian view we may have similar duties as well (Paez 2019 A Kantian Ethics of Paradise Engineering, Analysis). But we certainly would not have those duties on Kant's stated view.
(d) Line 64: it should say 'turn' rather than 'term'.
Author Response
(a) Reviewer: It is quite impossible for me to understand how (as the author says some authors suggest) 'Kant goes even further than the utilitarian would by formulating duties to insects' (106). Whatever Kantians say, if insects (or, more accurately, invertebrates in general) are sentient, then their interests must be factored in the utilitarian calculus. Moreover, more pragmatically, to the extent that we have evidence in support of invertebrate sentience, we are morally required to incorporate the possibility of harming/benefit them in our expected utility calculations. In fact, we do seem to have conclusive evidence of octopus sentience (see Cambridge Declaration on Consciousness, Low et al. 2012), and diverse degree of evidence regarding the sentience of other invertebrate animals. In addition, it would be interesting to discuss to what extent Kant's argument works for animals of whose lack of sentience we can be sure. If the indirect duty view necessitates the sentience of the entities regarding which we have the duties, then non-sentient animals would be excluded. If so, and if most invertebrates turn out not to be sentient, Kant's view and the utilitarian's would agree that we have no duties to them.
Comment (a): Thank you for bringing up this point. It is relevant, because Denis’ paper dates from 2000.
I would like to focus on insects only, because they are mentioned by Kant and because I see a difference to other invertebrates. While the sentience of cephalopods and crustaceans has already been acknowledged (e.g. in the Swiss Animal Protection Act in 2005) the debate regarding other invertebrates is still going on.
I have mentioned the actual debate about sentience and insects in an additional footnote 3: “Denis’ contribution [29] dates back to the year 2000. Since then, new scientific findings about the sentience of insects have given new impetus to the discussion [e.g. 65–67]. This means, if science can provide compelling reasons that insects are sentient, utilitarians would recognize them as morally relevant entities and Denis’ argument would be outdated.”
Reviewer: In addition, it would be interesting to discuss to what extent Kant's argument works for animals of whose lack of sentience we can be sure.
Comment: Kant does also consider non-sentient animals (and plants and non-animate nature) indirectly. But he makes a difference between sentient and non-sentient beings, because the danger of confusion of having direct duties (“Amphiboly in Moral Concepts of Reflection”) to sentient animals is higher regarding sentient animals.
(b) Reviewer: Regarding the argument from self-consciousness as differentia specifica, I think it would be gentler to the reader to (i) move the mention of it to a footnote and then (ii) provide a brief description of it.
Comment (b): I would like to leave it in the main text, but added a critical comment in an additional footnote 16:
“Whatever differentia specifica is mentioned by Kant, self-consciousness, rationality or moral autonomy (the last one is the most precise), it is an open question, if this inevitable has to result in a relationship of domination, where animals are no more regarded ‘as fellow creatures, but as means and instruments to be used at will for the attainment of whatever ends he [man] pleased.’ [45], 114. Instead of having a ‘prerogative which, by his [man’s] nature, he enjoyed over all the animals‘ [ibid.], the criterion could lead to a relation of compassion and care for the other creatures. Or a third relation could be an abolitionism, where humans should not interact with the other animals at all. It isn’t the place here to argue for one or the other position, but this consideration should make clear that Kant’s version of domination is neither the only human-animal relationship one can think of, nor is it self-evident.”
(c) Reviewer: There is a further problem in applied animal ethics which may help to show the steep practical divergence between Kant's ethics and utilitarianism. This is the problem of wild animal suffering. If, as the evidence suggests, wild animals tend to have lives of net suffering to naturogenic (as opposed to anthropogenic) harms, then we have utilitarian reasons to intervene in nature in order to help them. Even to the point of redesigning nature if necessary. Some have argued that on a modified Kantian view we may have similar duties as well (Paez 2019 A Kantian Ethics of Paradise Engineering, Analysis). But we certainly would not have those duties on Kant's stated view.
Comment (c): The topic of engineering wild animals is definitely interesting and worth to discuss. But I worry, that a (in-depth) discussion, that considers and explains the latest biotechnological innovations, would go beyond the scope of this paper. But I am looking forward to follow the discussion in my future work.
(d) Reviewer: Line 64: it should say 'turn' rather than 'term'.
Comment (d): The mistake is corrected. After the final reviews, the text (including all modifications and additional footnotes) will be proofread by a professional lecturer.
Reviewer 2 Report
In this paper three possible strategies to address Kant's denial of moral status to animals (MSA) are presented. One of these is then discussed in order to show that: a) its conclusions are less compatible with common sense morality than it is usually presented by its advocate; b) its conclusions are less compatible with animal rights and utilitarianism than it is usually presented by its advocate; c) it is an unsound position, both from a scientific and a normative standpoint. In the last part of the paper it is argued that the discussion of MSA in Kant is relevant to better understand his moral philosophy.
The view that is discussed in the paper is that Kant's treatment of MSA has (usually) overlooked far reaching consequences that make it an adequate approach in the context of the present normative attitudes toward animals. I believe the topic to be relevant. At first glance the conclusions of the paper may seem unappealing – they are, basically, a restatement of the limits of Kant's view on MSA. However, due to the various present attempts to push alternative readings on the issue, the article an its conclusions are well integrated in the debate.
Generally speaking, the Author does a good job in presenting Kant's moral philosophy in a detached while, at the same time, charitable perspective. The paper is a worthwhile addition to the literature and I recommend only minor revisions:
a) English. There are some typos (l. 62, treated; l. 64, linguistic term; l. 180 orphan parentheses – for instance) that should be corrected.
b) In the first sentence of the paper I am not sure that writing that “Historicaly, philosophers have shown little interest in the status of the human-animal relationship in Kant's ethics” is correct given that a few lines below Schopenhauer, Schweitzer and Nelson are mentioned as three examples of authors who criticized Kant on the subject.
c) The A. use the term “progressive” several times in the text in reference to Kant's view on MVA, but it is rather unclear to what sense the term is used. This is not surprising, since it is a rather vague word. The A. use it in order to qualify a desiderata for the advocates of the view that Kant's take on MVA has overlooked and far-reaching consequences – that is, “progressive”. However it is seem to be rather unclear how to interpret the word, and the risk here for the A. is to make a straw men of the adversaries. Usually I do not think that praising the quality of “being progressive” of a theory means much more than praising one's preferences. But “progressive” can be taken also as a synonym for “radical” or “innovative”. Is that the case? Does it mean here “radical”? But how can someone really claim that Kant's view, even if the really have overlooked and far-reaching consequences, could compete with – for instance – Regan's position, in term of “radicality”? Does it mean “innovative”? Yet, most of Kant's claims about animals had been already advanced in the past, and surely were not as much innovative as position endorsed, for instance, by classic authors like Plutarch. And so on. So, what is the content here of “being progressive”?
The last part of the paper is very interesting. However, it is also the part most vulnerable to critique due to its tackling with some of the most debated point in Kant's moral theory. On this regard, I highlight two issues that, in my humble opinion, need to be better expressed or clarified:
d) The A. writes that: “Even if Kant is right that persons with the capacity for autonomy are ends in themselves with absolute worth, it does not follow that everything else has only relative value for persons. However, this mantra is exactly what he repeatedly states throughout his work “ (l. 293-5). It may be a mantra, yet it is not fully clear to me how something cannot be and end for someone if it is not an end in itself. To me a position that accept a third moral category in-between end in itself and end for someone else sounds logically incongruous. This is not to say that Kant is right (that is, that animals have no moral standing). Yet, if one accepts his definition of end in itself, then, it seems to me that it follows that everything else has to be considered as an end for someone else, as the two concepts are contrary.
e) The A. does not tackle with the argument from self-consciousness differentia specifica because it would lead the discussion astray. I agree. However, in my view, this argument stands at the core of the famous quote of the Conjectures on the Beginnings of Human History about the sheepskin. Human beings recognize their place as ends of nature, according to Kant, at the end of a complex process in which they learn their emotional and cognitive difference from animals – that is, they learn to be/that they are creature of reason. It may then seem that the A. here is dismissing a core element to correctly represent Kant's idea of humanity as the end of nature – which, very probably, cannot be conflated with naive teleological positions à la Pangloss.
I would like to finish my review with a positive comment on this last section of the paper: I really appreciated the discussion of the argument reciprocity.
Author Response
General Comment: I would like to thank the reviewer for her/his careful reading, which made me rethink and clarify several points for myself!
a) Reviewer: English. There are some typos (l. 62, treated; l. 64, linguistic term; l. 180 orphan parentheses – for instance) that should be corrected.
Comment (a): The typos are corrected. After the final reviews, the text (including all modifications and additional footnotes) will be proofread by a professional lecturer.
b) Reviewer: In the first sentence of the paper I am not sure that writing that “Historicaly, philosophers have shown little interest in the status of the human-animal relationship in Kant's ethics” is correct given that a few lines below Schopenhauer, Schweitzer and Nelson are mentioned as three examples of authors who criticized Kant on the subject.
Comment (b): I specified and exchanged “philosophers” with “Kantians”. Side note: According to my knowledge this would make Nelson the first Kantian, who elaborated on the subject.
c) The A. use the term “progressive” several times in the text in reference to Kant's view on MVA, but it is rather unclear to what sense the term is used.
Comment (c): I use the term “progressive” always with regard to the historical context, in the sense of “ahead of its time”.
But to read it as “innovative” would also fit my understanding of Kant’s contribution. As Baranzke shows, Kants modification/rearrangement of the duties regarding animals from the the duties to others (lecture) to the duties to oneself (MST) is “a small revolution”.
The last part of the paper is very interesting. However, it is also the part most vulnerable to critique due to its tackling with some of the most debated point in Kant's moral theory. On this regard, I highlight two issues that, in my humble opinion, need to be better expressed or clarified:
d) The A. writes that: “Even if Kant is right that persons with the capacity for autonomy are ends in themselves with absolute worth, it does not follow that everything else has only relative value for However, this mantra is exactly what he repeatedly states throughout his work “ (l. 293-5). It may be a mantra, yet it is not fully clear to me how something cannot be and end for someone if it is not an end in itself. To me a position that accept a third moral category in-between end in itself and end for someone else sounds logically incongruous. This is not to say that Kant is right (that is, that animals have no moral standing). Yet, if one accepts his definition of end in itself, then, it seems to me that it follows that everything else has to be considered as an end for someone else, as the two concepts are contrary.
Comment: I try do give a twofold answer (but I am not sure, if I succeed). Unfortunately I am abroad right now and don’t have access to my literature. Please let me know, if the answer is not convincing. If needed, I will give in an additional answer within one week.
The logical problem exists only within a dualistic (normative) system (as Kant presupposes).
Regarding normativity: One could understand plants or ecosystems as end-in-it-selfes without any value for others (Taylor 1989: 71ff. was aware of this problem for his biocentrism).
Regarding dualism: I think even in Kant’s system it is possible to understand a sentient being as self-referential system with regard to pain and joy. If a sentient animal (or better several individuals of a population of wild animals) is in pain, the pain feels intrinsically bad for her. She values this experience as bad, without any reference to any other being or any person. This means, several wild animals exist as self-referential valuing systems, without having an absolute value, but also without having a relative value for persons.
In his book Logic (Log: A 23/IX 23f.) Kant writes that philosophy has an inherent, absolute value. Which is the definition of an end-in-itself. Because philosophy is not an entity that can confer an instrumental value to animals, at least one end-in-itself exists, that has no relation to animals. And animals have no relative value for this end-in-itself.
e) The A. does not tackle with the argument from self-consciousness differentia specifica because it would lead the discussion astray. I agree. However, in my view, this argument stands at the core of the famous quote of the Conjectures on the Beginnings of Human History about the sheepskin. Human beings recognize their place as ends of nature, according to Kant, at the end of a complex process in which they learn their emotional and cognitive difference from animals – that is, they learn to be/that they are creature of reason. It may then seem that the A. here is dismissing a core element to correctly represent Kant's idea of humanity as the end of nature – which, very probably, cannot be conflated with naive teleological positions à la
Comment (d): I really regret not to be able to discuss the subject of “self-consciousness” in-depth. I tried it in an earlier draft-version and had to dismiss it, because the discussion would need a paper of its own. I agree with the reviewer that it is a core element of humanity and I agree with Kant that there is a form of consciousness that cannot be found in any other species. But there are several problems with Kant’s argument:
- He speaks very general about self-consciousness and does not differentiate different forms of self-consciousness such as bodily self-awareness, social awareness and introspective awareness. According to current ethology, general denial of self-consciousness to all animals is problematic.
- The kind of self-consciousness that is required for Kant’s specific concept of autonomy needs to be defined more clearly. Kant needs to say that a highly sophisticated, second-order kind of self-awareness is a perquisite of acting autonomously. But this kind of self-awareness is not something that can be checked, or measured, with empirical methods. So, we must ask how Kant can deny it to all animals if he cannot present empirical evidence showing he is right to do so?
Side note: Interestingly, he mentions in unpublished notes from his lecture Physical Geography that elephants seem to be an analogon of morality (Analogon der Moralitaet, VPG Kaehler: 253), and that dogs are the most perfect animals because they show the ‘Analogon rationis’ most (VPG Kaehler: 401f.). But how does Kant know that those animals are only acting as if (moralanalog) they are moral beings?
- But the main problem of his argument is the conclusion, that ‘The fact that the human being can have the “I” in his representations raises him infinitely above all other living beings on earth’(Anth, 7: 127). Even if humans are the only being that are self-consciousness, how can he derive the conclusion of a “prerogative”, to use others “as means and instruments to whatever ends he pleases” (Conjectures on the Beginnings of Human History). One could also conclude that humans have a duty to care for all non-self-conscious beings or to leave them alone, because they are so different from us.
To conclude: Even if I cannot elaborate on a main feature of humanity I don’t think the “self-consciousness as differentia specifica argument” is a strong one. I added a critical comment in an additional footnote 16 that refers to the famous passage mentioned by the reviewer:
“Whatever differentia specifica is mentioned by Kant, self-consciousness, rationality or moral autonomy (the last one is the most precise), it is an open question, if this inevitable has to result in a relationship of domination, where animals are no more regarded ‘as fellow creatures, but as means and instruments to be used at will for the attainment of whatever ends he [man] pleased.’ [45], 114. Instead of having a ‘prerogative which, by his [man’s] nature, he enjoyed over all the animals‘ [ibid.], the criterion could lead to a relation of compassion and care for the other creatures. Or a third relation could be an abolitionism, where humans should not interact with the other animals at all. It isn’t the place here to argue for one or the other position, but this consideration should make clear that Kant’s version of domination is neither the only human-animal relationship one can think of, nor is it self-evident.”
Reviewer 3 Report
The authors objectives are clearly set out in the abstract and opening paragraph. The originality of the paper is in its detailed presentation of the gap between Kant's approach to animal ethics and the two mainstream approaches in rights theory and utilitarianism. Further is the detailed and critical examination of the 'indirect' account of ethical duties to animals as held by Kantians. This work is clearly upheld with examples and references to Kant's writings and clearly deserves publication. The paper is presented with due attention to clarity and meets relevant standards for publication in an English speaking journal. In fact I had only one minor correction.in line 34 colon should be followed with lower case. eg the.
Author Response
Comment: I would like to thank the reviewer for her/his encouraging comment. The spelling mistake has been corrected.
Reviewer 4 Report
In general, the Author's critics to Kant and to the Kantian commentators who argued in favor of Kant's argument for the indirect duties regarding animals do not add anything new to what has been already written against Kant's view on animals over the past decades (Pybus and Broadie 1974, Timmermann 2005, Ursula Wolf 2012, etc.). This is particular evident at pp. 4-5.
The views of Kant's "today's rival accounts" (4, 161) are portrayed only superficially with the intent to show their alleged superiority to the Kantian account. Unfortunately, in this way the reader cannot understand why several Kantian scholars find the Kantian alternative attractive when compared to Tom Regan's, Peter Singer's and Gary Francione's theories.
If, as the Author claims, Kant and the Kantian scholars fail "to provide a satisfactory argument showing why humans are allowed to use animals as mere means in general" (4, 155-156), he should discuss at least briefly how the rivals - Regan, Singer and Francione (only to mention the most popular theorists) -provide arguments for their views.
P. 4, 156: The Author mistakenly takes the instrumentalisation of animals for their mistreatment. He seems to forget also in the following analysis (pp. 6-7) that, according to Kant, there are several morally permissible ways to instrumentalise even humans.
P. 6, 277-278: The Author asks: "What are the arguments for the conclusion that animals are means to human ends, and that animals have only relative value for rational beings?". One could ask back (and this is in fact Denis's, Baranzke's, Kain's and Basaglia's point): What are the argument for the opposite conclusion, namely that animals have not only a relative, but an absolute value and their existence and interests are absolutely morally relevant? From a scientific point of view, there is no absolute value in nature: Not only animals, but also humans can be only means to each other. There is no scientific evidence that it is not allowed to instrumentalise other humans. The prohibition to instrumentalise other humans (or other animals) is not a scientific notion, but a normative one.
P. 7, 311-314. The Kingdom of Ends is presented by the Author as the foundation of Kant's notion of reciprocal obligation, which is if not mistaken, at least only partly correct. Each of us (humans) is subjected to the moral law not because we subject reciprocally each other, but because each of us (humans as Vernunftwesen - not only as rational beings) participate of pure practical reason, the only morally legislating instance. Otherwise, the distinction between homo houmenon and homo phenomenon (central for Kant's moral philosophy, ethics and also for his notion of dignity) would not be intelligible at all. The following parallel with the notion of "passive citizens’ in Kant’s political philosophy (315-334) as well as the Author's argument against Kant's view does not, in my opinion, hold.
Author Response
- 1. In general, the Author's critics to Kant and to the Kantian commentators who argued in favor of Kant's argument for the indirect duties regarding animals do not add anything new to what has been already written against Kant's view on animals over the past decades (Pybus and Broadie 1974, Timmermann 2005, Ursula Wolf 2012, etc.). This is particular evident at pp. 4-5.
Comment: I agree that the mentioned authors made some major contribution to the debate.
I don’t agree that the paper does not add anything new to the debate.
For example:
- The fact that animals occur at important sections and arguments in Kant’s opus has also not been mentioned by the authors referred to. This point is especially important and should be further developed in future research on the subject.
- The particular attempt of this paper is to explore and evaluate the argument of the far-reaching duties regarding animals in Kant’s ethics that is part of the actual debate, referring to literature that was published after the contributions of Broadie/Pybus 1974, Timmermann 2005 or Wolf 2012. None of these authors examine the contributions of Kain 2010, Altman 2014, 2018, Basaglia 2019, Camenzind 2018, 2020.
- None of the mentioned authors makes the distinction between the lecture notes on moral philosophy and the Metaphysics of Morals. But this distinction is relevant because some of the critique of Kant only concerns his lectures but not his critical writings. I intend here to introduce a higher degree of differentiation as it is often the case in the critique/defense of Kant’s ethic.
- The distinction between theory-immanent, theory-transcendent and defending strategies is the first time mentioned in this paper.
- The limitations of the argument of reciprocity are not mentioned by any of the above referred authors.
2. The views of Kant's "today's rival accounts" (4, 161) are portrayed only superficially with the intent to show their alleged superiority to the Kantian account. Unfortunately, in this way the reader cannot understand why several Kantian scholars find the Kantian alternative attractive when compared to Tom Regan's, Peter Singer's and Gary Francione's theories.
Comment: I am focusing on the reason that is prominently mentioned by several Kantians to defend Kant’s views attractive is the (alleged) far-reaching consequences regarding the treatment of animals in Kant’s ethic (e.g. Kain 2010, Altman 2014, 2018, Basaglia 2019). But there are other reasons, too.
I added a footnote (nr. 4) to overcome this shortage:
“Other reasons why Kant’s ethics is attractive for contemporary Kantians are summarized by Basaglia [cf. 31] (pp. 1725–1728). Among them are the strictness of Kant’s duties regarding animals (this point is further explored below in paragraph 2); the immunity of Kant’s ethics to the problems of utilitarianism and Schopenhauer’s ethics of compassion. The first one concerns the shortcoming, that the interests of a majority can trump the interests of a minority, and the second one the problem of universality of ethics of compassion. It states, that compassion cannot be the ground for a stable system of morality, because the capacity of compassion isn’t distributed equally and universally within humankind.”
It has to be further stated that the paper does not argue that Regans, Singers or Francione’s positions are superior to Kantians account in general, only regarding the consequences.
3. If, as the Author claims, Kant and the Kantian scholars fail "to provide a satisfactory argument showing why humans are allowed to use animals as mere means in general" (4, 155-156), he should discuss at least briefly how the rivals - Regan, Singer and Francione (only to mention the most popular theorists) -provide arguments for their views.
Comment: Thank you for bringing in this perspective. I understand that it may be dissatisfying for the reader that the paper does not give reason for an alternative position. But this endeavor cannot be made satisfying within this paper. All of the three mentioned positions differ significantly in their foundation (and also contradict each other).
I will provide a footnote (nr. 9) with literature, where the reader can find the foundations of alternative views:
“It would be beyond the scope of this paper to present reasons for the justification of the moral status of animals. Nevertheless, the reader will find in-depth inquiries in favor of the moral status of animals in the following works: Kantianism and animal rights: [8, 12, 15, 22, 26, 27, 47, 62]; Utilitarianism: [41]; Contractarianism [59, 60]; Virtue Ethics: [63, 64] and others that cannot be categorized clearly to the four traditions such as the capabilities approach or different biocentrist views: [6, 7, 38, 43, 57].”
4. P. 4, 156: The Author mistakenly takes the instrumentalisation of animals for their mistreatment. He seems to forget also in the following analysis (pp. 6-7) that, according to Kant, there are several morally permissible ways to instrumentalise even humans.
Comment: This must be a misunderstanding. The author is aware of the Kant’s distinction between “use as means” and “use as mere means” concerning humans. In the whole article the difference between “morally permissible instrumentalisation” and “morally impermissible/excessive instrumentalisation” is made explicit.
In the mentioned passages (p. 4, 156) the practices of hunting, animal experimentation and agriculture are morally impermissible forms of instrumentalization – based on the normative framework of the animal rights view – , because these practices violate animals’ right to bodily integrity, or their fundamental right to life.
5. P. 6, 277-278: The Author asks: "What are the arguments for the conclusion that animals are means to human ends, and that animals have only relative value for rational beings?". One could ask back (and this is in fact Denis's, Baranzke's, Kain's and Basaglia's point): What are the argument for the opposite conclusion, namely that animals have not only a relative, but an absolute value and their existence and interests are absolutely morally relevant? From a scientific point of view, there is no absolute value in nature: Not only animals, but also humans can be only means to each other. There is no scientific evidence that it is not allowed to instrumentalise other humans. The prohibition to instrumentalise other humans (or other animals) is not a scientific notion, but a normative one.
Comment: I agree with the reviewer, from a descriptive scientific theory one can neither defend nor promote a view, that animals have an absolute moral status.
The reference to Darwin is only used to state that Kant and Kantian’s cannot simply assume that animals are mere instruments for human ends. Darwin is not used to promote a position, that animals or humans should not be used as means at all. Neither does the paper argue that animals have an absolute, inherent value.
In this context paper does not aim to establish a normative position. The reference to Taylor and Korsgaard aim to show only that the Kantian position is not self-evident – as it is often presented within the defense of Kant’s indirect duty view. If the reader is interested how Korsgaard etc. argue for their position in-depth, he/she is invited to study their work. (see the newly added footnote 9)
6. P. 7, 311-314. The Kingdom of Ends is presented by the Author as the foundation of Kant's notion of reciprocal obligation, which is if not mistaken, at least only partly correct. Each of us (humans) is subjected to the moral law not because we subject reciprocally each other, but because each of us (humans as Vernunftwesen - not only as rational beings) participate of pure practical reason, the only morally legislating instance. Otherwise, the distinction between homo houmenon and homo phenomenon (central for Kant's moral philosophy, ethics and also for his notion of dignity) would not be intelligible at all. The following parallel with the notion of "passive citizens’ in Kant’s political philosophy (315-334) as well as the Author's argument against Kant's view does not, in my opinion, hold.
Comment: The paper does not say, that the “kingdom of ends” is the foundation of Kant’s notion of reciprocal obligation. The paper states that the symmetrical relations between persons create the kingdom of ends.
I understand the general difficulty to compare Kant’s political philosophy with his moral philosophy. But in this case I disagree with the reviewer. The use of Kant’s political philosophy is appropriate to illustrate that no axiology (regarding humans and animal) can be derived from the argument of reciprocity.
Further I don’t understand how the comment of the reviewer regarding the relation between homo noumenon and homo phenomenon provides reason against my argument. But this is interesting for the point I want to make: From the relation between subordinate (human being) and the legislating authority (homo noumenon), one cannot derive any value theory.
Reviewer 5 Report
The author provides a clear review of the prospects of a Kantian approach in animal ethics. His arguments are convincing. My only worry is about scope of this analysis. Since the author is concerned with Kantian and not only Kant’s arguments in animal ethics, more should be said about the different ways of being Kantian that have a relevance for animal ethics. Although the author’s review is complete with regards to the Kantian perspective as an individual ethics, it falls short of analyzing those approaches that are Kantian in a more political sense. Examples abound and not all are relevant for animal ethics: after all, the author mentions both Rawls’s and Habermas’s takes on this. But within the domain o post-Rawlsian scholarship something more should be said.
Indeed, the author discusses indirect views about animals as either Kantian or contractarian. But the picture is not complete because there also are political approaches, as discussed by Brian Berkey (“Prospects for an inclusive theory of justice: the case of non-human animals”, Journal of applied philosophy). The author gets close to this on p. 7 when he discusses the argument from reciprocity. However, he does not consider the whole spectrum of the debate. A fuller reconstruction should also include Angie Pepper (“Political Liberalism, Human Cultures, and Nonhuman Lives”), Federico Zuolo (“The priority of suffering over life. How to accommodate animal welfare and religious slaughter”, The Ethics Forum). See also, Robert Garner, “Rawls, animals and Justice: new literature, same response”, Res publica.
Finally, I was surprised not to see what I found the most convincing argument showing the incoherence in Kant’s position about animals, which has been provided by Cavalieri, P. (2001) The Animal Question. Why Nonhuman Animals Deserve Human Rights (New York: Oxford University Press).
Author Response
Comment: Thank you very much for taking the time to read my manuscript. I am delighted to read that you think that arguments are convincing.
1. I appreciate the additional literature you mentioned. I was aware of the political work of Nussbaum, Donaldson/Kymlicka, Garner, O’Sullivan or Cochrane, but not Berkey, Pepper or Zuolo. I would prefer to stay within the field of moral philosophy, because the “argument of the far-reaching duties regarding animals” is discussed in this context. Nevertheless, to mention the political turn in animal ethics,
I added a footnote (nr. 3) with relevant literature:
“In this paper it won’t be possible to discuss approaches that were developed to improve Rawls’ and Habermas’ short-comings regarding the moral status of animals. Nevertheless, several contemporary works discuss this issue [68–72].
I added the following authors [68–72].:
Berkey B. (2017) ‘Prospects for an inclusive theory of justice: the case of non-human animals’ Journal of Applied Philosophy, 34 (5), pp. 679–695.
Cochrane, A. (2010) An Introduction to Animals and Political Theory, New York: Palgrave Macmillan.
Cochrane, A. (2018) Sentientist Politics. A Theory of Global Inter-Species Justice, Oxford: Oxford University Press.
Nussbaum, M. C. (2007) Frontiers of Justice: Disability, Nationality, Species Membership, Cambridge et al.: Harvard University Press.
Garner, R. (2013) A Theory of Justice for Animals. Animal Rights in a Nonideal World, Oxford: Oxford University Press.
2. I was delighted to read Cavalieris’ contribution. I don’t know, why she is not quoted by Kantians more often. Without explicit mentioning the issue, her work supports my conclusion that the animal question is related to the core of Kant’s philosophy.
I added the following footnote 18:
“Without explicit mentioning the issue, Cavalieris’ work supports my conclusion that the animal question is related to the core of Kant’s philosophy [cf. 73], (pp. 48–58).”
General remark: After the final reviews, the text (including all modifications and additional footnotes) will be proofread by a professional lecturer.
Round 2
Reviewer 4 Report
See please my previous review.
Author Response
Please see the attached revised version
